# Aspirin Hypersensitivity in Patients with Coronary Artery Disease: An Updated Review and Practical Recommendations

**DOI:** 10.3390/biom14101329

**Published:** 2024-10-19

**Authors:** Luigi Cappannoli, Stefania Colantuono, Francesco Maria Animati, Francesco Fracassi, Mattia Galli, Cristina Aurigemma, Enrico Romagnoli, Rocco Antonio Montone, Mattia Lunardi, Lazzaro Paraggio, Carolina Ierardi, Ilaria Baglivo, Cristiano Caruso, Carlo Trani, Francesco Burzotta

**Affiliations:** 1Dipartimento di Scienze Cardiovascolari—CUORE, Fondazione Policlinico Universitario A. Gemelli IRCCS, 00168 Rome, Italy; 2Facoltà di Medicina e Chirurgia, Dipartimento di Scienze Cardiovascolari—CUORE, Università Cattolica del Sacro Cuore, 20123 Rome, Italy; 3UOSD DH Medicina Interna e Malattie Dell’apparato Digerente, Fondazione Policlinico Universitario A. Gemelli IRCCS, 00168 Rome, Italy; 4Maria Cecilia Hospital, GVM Care & Research, 48033 Cotignola, Italy; 5Department of Medical-Surgical Sciences and Biotechnologies, Sapienza University of Rome, 00185 Latina, Italy; 6UOC CEMAD Dipartimento di Scienze Mediche e Chirurgiche Addominali ed Endocrino Metaboliche, Fondazione Policlinico Universitario A. Gemelli IRCCS, 00168 Rome, Italy; 7UOSD Allergologia e Immunologia Clinica, Dipartimento Scienze Mediche e Chirurgiche Addominali ed Endocrino Metaboliche, Fondazione Policlinico Universitario A. Gemelli IRCCS, 00168 Rome, Italy

**Keywords:** aspirin hypersensitivity, coronary artery disease, desensitization, low-dose ASA challenge

## Abstract

Acetylsalicylic acid (ASA) represents a cornerstone of antiplatelet therapy for the treatment of atherosclerotic coronary artery disease (CAD). ASA is in fact indicated in case of an acute coronary syndrome or after a percutaneous coronary intervention with stent implantation. Aspirin hypersensitivity is frequently reported by patients, and this challenging situation requires a careful evaluation of the true nature of the presumed sensitivity and of its mechanisms, as well as to differentiate it from a more frequent (and more easily manageable) aspirin intolerance. Two main strategies are available to allow ASA administration for patients with CAD and suspected ASA hypersensitivity: a low-dose ASA challenge, aimed at assessing the tolerability of ASA at the antiplatelet dose of 100 mg, and desensitization, a therapeutic procedure which aims to induce tolerance to ASA. For those patients who cannot undergo ASA challenge and desensitization due to previous serious adverse reactions, or for those in whom desensitization was unsuccessful, a number of further alternative strategies are available, even if these have not been validated and approved by guidelines. The aim of this state-of-the-art review is therefore to summarize the established evidence regarding pathophysiology, clinical presentation, diagnosis, and management of aspirin hypersensitivity and to provide a practical guide for cardiologists (and clinicians) who have to face the not uncommon situation of a patient with concomitant coronary artery disease and aspirin hypersensitivity.

## 1. Introduction

Acetylsalicylic acid (ASA, aspirin) represents a cornerstone of antiplatelet therapy for prevention and treatment of atherosclerotic coronary artery disease (CAD) [1]. Since 1899, when it was first synthetized, its use has largely spread as an anti-inflammatory, pain-relieving drug. Once its antiplatelet effects and effectiveness in preventing atherothrombosis at low doses were discovered, it quickly became one of the most widely used drugs worldwide for patients with ischemic heart disease [2]. Aspirin (at a lower than anti-inflammatory dose) is indicated in patients with acute coronary syndromes (ACSs) and in those who undergo percutaneous coronary interventions (PCIs) with stent implantation, as well as in primary prevention for those with very high risk of myocardial infarction (MI) [3,4]. ASA also finds application in peripheral artery disease and cerebrovascular disease, and it is also indicated in other non-cardiovascular diseases (rheumatological diseases and cancer) [5]. As an antiplatelet agent (75–100 mg/daily as a maintenance dose) it can be used alone or as part of a dual/triple antithrombotic therapy (in combination with anticoagulants and/or other antiplatelets agents), depending on clinical presentation, and usually its assumption is prolonged for at least 6–12 months, but more often for the patient’s entire life [1,6].

Given this wide range of indication and its widespread utilization in everyday clinical settings, it is not rare that a patient reports that he cannot consume aspirin or other non-steroidal anti-inflammatory drugs (NSAIDs) due to a hypersensitivity or intolerance. This turns to be a very challenging situation that requires a careful evaluation, first of the true nature of this presumed hypersensitivity, and then of its mechanisms and clinical manifestations. Patients may, in fact, report as drug hypersensitivity common side effects related to aspirin and other NSAIDs (such as retrosternal pyrosis and other gastrointestinal manifestations) and, more importantly, may be unable to provide a detailed description of the presentation and the drugs related to the presumed hypersensitivity [7,8].

The aim of this state-of-the-art review is therefore to summarize the established evidence regarding pathophysiology, prevalence, diagnosis, and management of aspirin hypersensitivity and to provide a practical guide for everyday clinical practice for cardiologists (and clinicians) who face the not so uncommon situation of a patient with concomitant CAD and reported aspirin hypersensitivity.

## 2. Definition

Aspirin (and NSAIDs in general) hypersensitivity can be defined as any unpredictable adverse drug reaction occurring in susceptible patients. Any adverse reaction that can be clinically expected, if the drug is given at sufficient dose and based on its pharmacodynamic, is not to be considered a drug hypersensitivity [9]. The mechanisms and clinical manifestations of aspirin and NSAID hypersensitivity are similar, as are their diagnostic and therapeutic approaches, thus from now on in this text the term “aspirin hypersensitivity” will be used to refer to both aspirin and NSAID hypersensitivity (if not otherwise specified). Similarly, the terms “sensitivity” and “hypersensitivity” have been previously used as synonyms, and from now on the term hypersensitivity will be preferred. On the contrary, the term “aspirin intolerance” refers to a number of signs and symptoms (such as retrosternal pyrosis, dyspepsia, bleeding, etc.) related to the pharmacodynamic of the drug, manifesting in susceptible subjects and not to be considered as hypersensitivity reactions.

Aspirin hypersensitivity can be classified based on its pathophysiological mechanism into nonallergic (non-immunologically mediated, based on a pharmacological mechanism) and allergic (immunologically mediated) [10]. While the former may be dose-dependent, depending on the COX-1 inhibition, the latter are usually dose-independent, so the mechanism described in those cases applies to both 75 mg and 100 mg doses.

Lastly, as per other NSAIDs, aspirin hypersensitivity can manifest by different signs and symptoms, but the main clinical presentations are represented by the following five conditions: aspirin/NSAID-exacerbated respiratory tract disease (NERD), aspirin/NSAID-exacerbated cutaneous disease (NECD), aspirin/NSAIDs-induced urticaria/angioedema (NIUA), selective aspirin/NSAIDs-induced urticaria, angioedema, and/or anaphylaxis (SNIUAA), and selective aspirin/NSAIDs-induced delayed reactions (SNIDR) [11] (Table 1).

## 3. Pathophysiology

### 3.1. Cyclooxygenases Inhibition

NSAIDs inhibit the enzyme named prostaglandin G/H synthase, commonly known as cyclooxygenase (COX), which is involved in the metabolism of the arachidonic acid (AA) that leads to prostanoid formation (Figure 1). Two different COX isoforms exist, COX-1 (ubiquitarian among various tissues) and COX-2 (which is synthetized in response to cytokine stimuli during inflammation and other pathological process such as cancer) [12]. COXs catabolize the formation of prostaglandin (PG) G2 from AA, and then a hydroperoxidase transforms PGG2 into PGH2; from the latter, different isomerases and synthases lead to the formation of terminal prostanoids, including different PGs, thromboxane (Tx), and prostacyclin (PGI2) [13]. Aspirin, among all traditional NSAIDs, has the peculiarity to inhibit both COX-1 and COX-2 enzymes permanently and irreversibly. Therefore, the duration of the aspirin effect depends only on the turnover velocity of new enzyme formation, which varies among different cells.

Prostanoids have different effects among body tissues and systems. PGE2, PGI2, and PGD2 promote vasodilation and inhibit platelet aggregation, whilst TxA2 promotes vasoconstriction and is the main mediator of platelet aggregation [14]. Platelets are anucleate cells containing only the enzyme isoform COX-1, without protein synthesis capacity. Therefore, after aspirin-induced permanent inhibition of COX-1 in circulating platelets, their TxA2 production is inhibited, as well as their aggregation, until new platelets are replaced. PGs, and in particular PGE2, are also responsible for fever induction and for pain receptors hypersensitivity increase (PGE2 and PGI2 reduce the receptors’ stimulation threshold), inducing hyperalgesia and allodynia [15]. PGs are also key mediators in inflammatory processes. PGI2 and PGD2 have opposite functions: the former promotes inflammation; the latter is the main anti-inflammatory PG. PGE2 elicits its action by different E prostanoid receptors (EP) and has opposing functions (pro- or anti-inflammatory) based on the level and type of EP expression by a single cell and tissue. Among all PGs, PGE2 has a pivotal role in aspirin hypersensitivity and, in particular, it has protective effects through EP2 against the symptoms of asthma [16], and counteracts leukotriene-mediated bronchoconstriction by inducing smooth muscle cells relaxation [17]. The PGE2/EP2 axis has also other beneficial effects, such as inhibiting eosinophil migration and mast cells degranulation, leading to decreased circulating levels of histamine and tryptase [18,19]. Lastly, PGE2 inhibits 5-lipoxygenase (5-LO) migration (and therefore activity) toward the nuclear membrane, which leads to a reduced production of cysteinyl leukotrienes, key mediators of inflammation and allergic reactions [20]. In patients with NERD, there is evidence of both reduction in PGE2 level and in EP2 receptors expression [21].

Based on its pathophysiological mechanisms, aspirin hypersensitivity can be classified into nonallergic (non-immunologically mediated) and allergic (immunologically mediated) reactions [10].

### 3.2. Nonallergic (Pharmacological) Reactions

Nonallergic reactions include non-immunologically mediated reactions, depending only on inhibition of the COX-1 pathway. COX-2 inhibitors, in fact, do not induce this kind of reactions [22].

Due to COX-1 inhibition, PGE2 downstream production is reduced as well as its favorable actions. This, in subjects with an intrinsic susceptivity due to an alteration in arachidonic acid metabolism, turns into a 5-LO pathway hyperactivation and leukotrienes production, in particular leukotrienes E4 and D4, which are responsible for inflammatory response typical of asthma and histamine release from mast cells [23,24].

Patients with this kind of aspirin hypersensitivity are particularly susceptible to the effects of leukotrienes and, usually, will react also to other NSAIDs (due to the non-immunological nature of these reactions, based only on pharmacological inhibition of COX-1 enzyme). These patients are usually referred to as cross-reactors, and the hypersensitivity as cross-intolerance [11].

The reactions included in this group are: NERD, NECD, and NIUA.

It is also to be noted that there are hypersensitivity reactions (HRs) to foods in which NSAIDs act as aggravating factors (NSAID-exacerbated food allergy [NEFA]) or cofactors (NSAID-induced food allergy [NIFA]), often misdiagnosed as HRs to NSAIDs. In this context, this condition is anything but negligible. A recent study on 252 patients previously diagnosed with NSAID hypersensitivity demonstrated that almost 40% of them tolerated drug challenge with the suspected NSAID, excluding the diagnosis of HR to NSAIDs. In these cases, the main responsible food allergen was a lipid transfer protein, Prup3 [25]. Non-specific lipid transfer proteins (nsLTPs) are a family of plant pan-allergens that represent the primary cause of food allergies in the Mediterranean area, characterized by a wide range of clinical manifestations, ranging from the total absence of symptoms up to anaphylaxis [25]. nsLTPs are widely present in fruits belonging to Rosaceae family, especially *peach*, but also *apricot*, *apple*, *pear*, and many other stone fruits, as well as nuts (*walnut*, *hazelnut*, *cashew nut*, and *peanut*) and other vegetables (*wheat*, *tomato*, *lettuce*, *maize*, *green bean*, *goji berry*, *eggplant*, *sunflower seed*, and *flaxseed*). In LTP-sensitized subjects, NSAIDs may act as co-factor, promoting adverse reactions, so allergist consultation is essential for diagnostic workout, in order to evaluate food allergy in those cases. An accurate dietary anamnesis about all ingested foods within 4 h before and after NSAID exposure, skin prick tests with commercial extracts or fresh food, and assays of serum specific IgE and molecular allergens will be useful to confirm diagnosis.

### 3.3. Allergic (Immunological) Reactions

Allergic reactions include immunologically mediated reactions, depending on drug-specific IgE production against aspirin or on a specific T-cell response.

In this case, aspirin (or another NSAID) acts as a hapten, inducing specific IgE production, and therefore an immediate immune reaction (after further expositions) with mast cell degranulation and histamine release due to drug-specific IgE antibody production and circulation. The mechanisms of delayed reactions are less clear, and probably involve T-cell specific response [11].

Patients with this kind of aspirin hypersensitivity usually will not react to other NSAIDs (due to the drug specific immunologically mediated response) and they are usually referred also as selective reactors [11].

The reactions included in this group are: SNIUAA and SNIDR.

## 4. Prevalence

The exact prevalence of aspirin hypersensitivity is difficult to establish, due to previous studies reporting contrasting results and to different diagnostic criteria used. Moreover, aspirin hypersensitivity includes a variety of different clinical entities, each one with its own prevalence. The estimated overall prevalence of aspirin hypersensitivity varies from 0.6% to 2.5% in general population and from 4.3% to 11% in asthmatic patients [26,27,28].

In particular, aspirin-exacerbated respiratory (and cutaneous) disease usually affects patients from the fourth decade of life, but it remains often under-diagnosed because patients (and physicians) do not attribute the symptoms to aspirin assumption [29]. Previous data reported a wide range of prevalence of aspirin-exacerbated respiratory tract disease (from 1.2% to 44%, depending on the population studied and the method of the diagnosis) [30], and a recent meta-analysis by Rajan et al. reported that the prevalence for such a condition was 7.15% on average in young adult suffering from asthma (reaching a 14.9% among patients with severe asthma), 9.7% in those with nasal polyps, and 8.7% in those with chronic rhinosinusitis [31].

The reported prevalence of aspirin-induced urticaria/angioedema varies among general population from 0.07% to 0.2% [10]. The exact prevalence of aspirin-induced delayed reaction is not known.

## 5. Clinical Presentation

The main five clinical presentations of aspirin hypersensitivity are NERD, NECD, and NIUA for the non-immunologically mediated reactions, and SNIUAA and SNIDR for the immunologically mediated reactions (Table 1).

### 5.1. Aspirin/NSAID-Exacerbated Respiratory Disease (NERD)

This form of aspirin hypersensitivity is caused by the excessive production of leukotrienes, causing asthmatic symptoms and naso-ocular secretions 1–3 h after drug assumption. The natural history of the disease is usually preceded by chronic upper airways disease (chronic rhinosinusitis, nasal polyposis, asthma) [32].

The clinical presentation includes bronchoconstriction with dyspnea and wheezing, rhinorrhea, and nasal congestion. This is the most frequent and well-known form of aspirin hypersensitivity, and it has been previously named as aspirin/Samter’s triad (characterized by aspirin intolerance, asthma, and nasal polyposis), Widal’s syndrome (the first one reporting this condition) [33], aspirin-induced asthma, or aspirin-sensitive rhinosinusitis/asthma syndrome. In the most severe forms, patients may experience laryngeal angioedema and can even require intubation and ventilatory support [9].

Patients with this form of aspirin hypersensitivity will usually react also to other NSAIDs (as the mechanism is not drug-specific) and are good responder to ASA desensitization [10,34].

### 5.2. Aspirin/NSAID-Exacerbated Cutaneous Disease (NECD)

This form of aspirin hypersensitivity occurs in patients with history of chronic spontaneous (idiopathic) urticaria, with symptoms exacerbated by aspirin exposition in 15–25% of cases, and manifesting with typical urticaria signs and/or angioedema by one hour from drug assumption [35,36]. The skin manifestations usually involve the head (eyelid and lip) and extremities, but can also affect the upper airways, with symptoms similar to NERD in case of laryngeal angioedema [36]. The reaction persists for hours but, in some cases, may last for several days [37].

Patients with this form of hypersensitivity cross-react with other NSAIDs and usually do not respond to drug desensitization, experiencing frequent urticaria flare up during treatment [34].

### 5.3. Aspirin/NSAIDs-Induced Urticaria/Angioedema (NIUA)

The clinical presentation of this form of hypersensitivity is similar to that of NECD, but it occurs in patients without a previous history of urticaria/angioedema. Even in this case patients cross-react with other NSAIDs, but drug desensitization can be usually achieved [10,34].

### 5.4. Selective Aspirin/NSAIDs-Induced Urticaria, Angioedema or Anaphylaxis (SNIUAA)

These include classic IgE-mediated allergic reactions. The symptoms onset is within a few minutes from exposition (generally by the first hour) and, as with other allergens, range from mild pruritis, skin swelling, and urticaria up to severe reactions including diffused angioedema, laryngeal obstruction, and anaphylaxis with respiratory involvement (wheezing, dyspnea, and bronchospasm) [9,38].

Patients with this form of aspirin hypersensitivity do not cross-react with other NSAIDs (because the pathological mechanism is IgE specific) and do not have clinical history of previous chronic respiratory diseases or urticaria/angioedema. Desensitization can be achieved in these patients [10].

### 5.5. Selective Aspirin/NSAIDs-Induced Delayed Reactions (SNIDR)

The exact pathogenesis and the underlying mechanisms of these kind of reactions are not completely known, but they have been attributed to a specific T-lymphocyte immunological reaction, with symptoms onset within 24–48 h after drug exposition [39]. Those include a variety of skin manifestations, such as photohypersensitivity, delayed urticaria, and contact dermatitis, up to more severe forms such as drug reaction with eosinophilia and systemic symptoms (DRESS) or Stevens–Johnson syndromes and toxic epidermal necrolysis [40,41].

Aspirin is less frequently involved in this kind of hypersensitivity compared to other NSAIDs (including COX-2 inhibitors) and patients do not cross-react with other drugs [42].

## 6. Diagnosis

The diagnostic criteria for aspirin hypersensitivity (and NSAID hypersensitivity in general) have been previously described in position papers and updated guidelines [9,11,43].

A precise interrogation of patients with suspected aspirin hypersensitivity with detailed clinical history is mandatory. Moreover, patients with cutaneous and/or anaphylactic reactions should be carefully questioned about all foods ingested within 4 h before or after NSAID exposure, and targeted food allergy tests should be considered in the diagnostic workup, to exclude LTP sensitization [25]. Patients with aspirin hypersensitivity usually report episodes of rhinorrhea, dyspnea, and other upper airways symptoms within one or two hours after aspirin assumption. Alternatively (or in addition to those symptoms), patients could also report cutaneous reactions or present lower airways manifestations (i.e., bronchospasm) [44]. When clinical history is suggestive for aspirin hypersensitivity, patients should be evaluated with further investigations, to confirm the hypersensitivity and the specific pathogenetic mechanism.

Provocation tests are recognized as the gold standard for aspirin hypersensitivity diagnosis and different protocols have been proposed. These include: oral aspirin challenge, bronchial (inhalation) L-lysine-aspirin (L-ASA) challenge, and nasal L-ASA challenge [43].

The oral full-dose aspirin challenge is the more sensitive and specific test for diagnosing ASA hypersensitivity [45,46,47]. This procedure is contraindicated in subjects with previously reported severe reactions to the specific drug and should be performed by experienced operators [9]. Forced expiratory volume in 1 s (FEV1) is measured at baseline (before exposition). Incremental doses of aspirin (27, 44, 117, 312 mg) are administered per os at 90–120 min intervals (up to a cumulative dose of 500 mg). In case of very high suspicion of aspirin hypersensitivity, a further dose of 500 mg (up to a cumulative dose of 1000 mg) can be administered. In case of safety concerns, the first dose can be divided in two. FEV1 is measured immediately after each dose and then every 30 min. A decrease in FEV1 > 20% is diagnostic. Even if such reduction is not reached, but typical severe extrabronchial symptoms (rhinorrhea, nasal congestion, ocular injection, periorbital swelling, or urticaria/angioedema) are reproduced, the test is considered positive [43,44].

The bronchial (inhalation) L-ASA challenge requires less time to be performed than the oral test, and it has a lower incidence of adverse reactions, although it is slightly less sensitive [46,48]. The general considerations for this test are similar to the oral challenge. The drug is administered in the form of crystalline lysine-aspirin diluted in saline at different concentrations and then nebulized. The patient inhales the drug in incremental doses (obtained varying the number of breaths from the nebulizer and the concentration of the solution) every 30 min, up to a 181.98 mg cumulative dose. FEV1 is measured every 10 min. The diagnostic criteria are the same as for the oral challenge [43,44].

The nasal provocation test is less sensitive than the previous ones and a negative result does not rule out the diagnosis. It is mainly used for patients with contraindications for oral or inhalation tests and for those with predominantly nasal symptoms (such as chronic rhinosinusitis and polyposis) [49,50]. The parameters collected for this test include clinical symptoms, total nasal flow (measure by acoustic rhinometry), and/or inspiratory nasal flow (measured by active anterior rhinomanometry or peak nasal inspiratory flow). L-ASA is prepared in the same way as in the inhalation test and a dose of 80 μL (for a total of 16 mg) is instilled by a pipette into each nostril in one minute. Then, nasal symptoms, inspiratory flow, and nasal volumes are measured every 10 min for the following two hours. The test is positive in case of nasal symptoms occurrence and 25% decrease of total nasal flow or 40% bilateral drop of inspiratory nasal flow [43,44].

Other diagnostic methods have been proposed to detect various forms of aspirin hypersensitivity, and those include in vivo skin tests and some laboratory tests (such as the cellular allergen stimulation test and the basophil activation test and the dosage of LTE4 urinary levels), but their specificity and hypersensitivity are lower and are currently not recommended as the standard for diagnosing aspirin hypersensitivity [11,44]

## 7. Management of Aspirin Hypersensitivity in Patients with Coronary Artery Disease

According to recent evidence, two methods are available for administering ASA antiplatelet therapy to patients with atherosclerotic cardiovascular disease and a history of hypersensitivity reactions (HRs) to ASA or two or more different NSAIDs: a low-dose ASA challenge (LDAC)—a diagnostic test aimed to assess the tolerability of ASA at an antiplatelet dose of 100 mg—and ASA desensitization, a therapeutic procedure which aims to induce tolerance to ASA [51,52,53]. While LDAC is recommended for stable settings, ASA desensitization is the preferred choice for patients with ACS [54].

In a study by Cortellini et al., the authors investigated both protocols (ASA desensitization and LDAC) on a group of 103 patients which reported HRs to ASA and/or other NSAIDs: 82 patients underwent LDAC, while 21 patients underwent ASA desensitization [51].

### 7.1. Low-Dose ASA Challenge (LDAC)

In the study by Cortellini et al., the authors proved that LDAC at a cumulative dose of 110 mg may be a simple, reliable, and cost-effective procedure to allow a low-dose ASA administration in stable settings for patients with coronary artery disease and aspirin hypersensitivity. LDAC was conducted in an outpatient clinic, administering increasing doses of 10, 25, 25, and 50 mg of ASA at 45 min intervals, reaching a cumulative dose of 110 mg. The test was considered positive, and stopped, if any of the following occurred: cutaneous (erythema, urticaria, and/or angioedema) or respiratory reactions (including rhinorrhea, nasal obstruction, sneezing, cough, and dyspnea with a decrease of ≥ 20% in the FEV1), and hypotension, with or without gastrointestinal symptoms. Patients were discharged one hour after the last ASA administration and were advised to report any further signs or symptoms of reactions to ASA within the following 48–72 h. The test resulted positive in two patients (3.2%) [51]. These results are in accordance with another previous study by the same authors [54].

LDAC is therefore an effective diagnostic test to identify those patients with presumed history of HRs to ASA who tolerate the drug at antiplatelet dose and to select those few “true” ASA-sensitive patients who may benefit from desensitization. In this way, desensitization procedures—which are expensive and more difficult to execute—can be spared in most patients who report HR to ASA. Moreover, thanks to LDAC, a dose of ASA 75 mg could be taken into account in patients with ASA hypersensitivity who cannot tolerate a greater (i.e., 100 mg or more) dose. Lastly, LDAC represents the strategy of choice in chronic and stable patients, whilst ASA desensitization is to be preferred in acute settings.

### 7.2. Aspirin Desensitization

Desensitization protocols consist of administering multiple and progressive dose increases of drugs to which patients are known to be hypersensitive [53]. The aim of those protocols is to mitigate the sensitivity to the molecule responsible for the adverse reactions and to make it possible for patients to take it without any risk. The most recent aspirin desensitization protocols in patient with coronary artery disease induce tolerance to aspirin dosages up to 100 mg or even more, with effectiveness around 100% (and the same efficacy is obviously expected for the 75 mg dose) [7].

Classic desensitization protocols often need several days to be completed, making them cumbersome [55]. A recent review by Verdoia et al. showed that at least 21 different desensitization protocols, published between years 2000 and 2023 for a total population of over 1000 desensitized patients, were safe and efficient in achieving tolerance to ASA, with no significant distinctions found between intravenous and oral administration and without relevant differences between desensitization protocols which included a pre-medication therapy (for example, corticosteroids, antihistamine, antileukotrienes, or a combination of these) and protocols that did not include it [56]. A relatively recent meta-analysis of 15 studies, for a total of 480 ACS patients with ASA HRs, reported a success rate of 98.3% for different desensitization protocols [57].

Among rapid desensitization protocols, Rossini et al. reported the greatest sample size (330 patients) and the best efficacy and safety data [52]. In this study, a rapid desensitization protocol was performed (within 5.5 h), in contrast with the classic bulky desensitization methods. ASA desensitization was carried out before coronary angiographies in all patients included in the study, except for those presenting STE-ACS or, in general, in urgent/emergency settings, when desensitization protocol was postponed after percutaneous coronary intervention. Six sequential doses of aspirin (1, 5, 10, 20, 40, and 100 mg) were orally administered within 5.5 h (at minutes 0, 30, 60, 90, 210, and 330, respectively). Meanwhile, the operator measured pulse, saturation, and blood pressure every 30 min. Nasal, ocular, respiratory and mucocutaneous reactions were also strictly observed up to four hours after the end of the protocol and, if any adverse event occurred, ASA administration was immediately interrupted. After desensitization, every patient continued a daily oral assumption of ASA to avoid the recurrence of adverse effects shortly after discontinuation. In all elective cases, the use of antihistamines, steroids, and antileukotrienes was stopped for a week before desensitization trial. The desensitization procedure was successful in 95.4% of cases (315 out of 330 patients).

In the aforementioned study by Cortellini et al., the authors reported similar results, with a 100% success rate regarding their rapid ASA desensitization protocol [51]. In this study, the desensitization protocol was performed administering 0.1, 1, 2, 3, 4, 5, and 10 mg of ASA every 20 min, then 15 mg after 40 min, 25 mg after 1 h, and 35 mg after another hour, reaching the cumulative dose of 100.1 mg. The authors reported that desensitization was successful in 100% of patients.

The results of these studies proved that a fast desensitization protocol is both safe and efficient for patients with aspirin hypersensitivity who are undergoing coronary angiography, regardless of the nature of the sensitivity. Nevertheless, ASA desensitization still remains a long and expensive process, often difficult to be carried out in emergency situations [58]. In addition to this, Verdoia et al. reported that protocols involving more than six escalating doses demonstrated higher success rates compared to those with six or fewer doses (99.2%, CI [97.9% to 100.4%] vs. 95.4%, CI [93% to 97.8%]; p = 0.007), highlighting the potential benefits of a slower escalation of ASA therapy, which adds another difficulty to face during emergency situations such as ACS [56].

In conclusion, patients at high risk of or with established atherosclerotic cardiovascular disease and a suspected history of ASA hypersensitivity should first undergo an ASA challenge, if the clinical setting allows this approach. In case of positive result of an ASA challenge, documented history of hypersensitivity to aspirin at a dose equal to or less than 100 mg, or in case of acute coronary syndromes, ASA desensitization is to be preferred [54,59]. Figure 2 (central figure) provides a practical algorithm to guide the decision-making in case of CAD in a patient reporting ASA hypersensitivity, depending on the clinical setting.

### 7.3. Alternative Strategies to Overcome Aspirin Hypersensitivity

For patients requiring DAPT and with previous severe HR to aspirin or anaphylaxis, both LDAC and desensitization are contraindicated. In this case, or for those patients who underwent unsuccessful desensitization, a number of possible alternative strategies have been proposed, even if none have been validated and recognized by clinical guidelines.

#### 7.3.1. Indobufen

Different studies suggested the possibility to replace aspirin with other antiplatelet agents, based on similarities of pharmacodynamic properties. Indobufen, an isoindolinyl phenyl-butyric acid derivative, reversibly inhibits platelet aggregation by suppressing thromboxane A2 production acting on COX-1. A recent trial compared a DAPT regimen based on indobufen (100 mg twice a day) plus clopidogrel vs. standard DAPT (ASA + clopidogrel) in more than 4000 Chinese patients undergoing PCI, reporting better clinical outcomes in the indobufen group (a significant reduction in bleeding events without an increase in ischemic events) [60]. The phase 4 NCT05105750 trial is currently ongoing, and it aims at comparing indobufen 200 mg twice a day versus aspirin 100 mg daily in patients with established atherosclerotic coronary artery disease. The results of this study may provide further insights about the possibility of replacing aspirin with indobufen in patients with ASA hypersensitivity.

#### 7.3.2. Dipyridamole

Dipyridamole is an old antiplatelet and coronary vasodilator agent that inhibits platelet phosphodiesterase and increases interstitial adenosine levels. Previous animal studies demonstrated the synergistic antithrombotic effects of dipyridamole and aspirin in rabbits, but these have not been demonstrated in human trials yet. The PARIS (Persantine-Aspirin ReInfarction Study) I and II trials (performed in 1980 and 1986, respectively) failed to assess the superiority of aspirin plus dipyridamole compared to aspirin alone in secondary prevention in reducing the risk total mortality, cardiovascular mortality, and myocardial infarction [61]. This hypothesized synergy has not been demonstrated even with clopidogrel in a nationwide case–control study, probably due to a similar pharmacodynamic effects of both molecules [62].

#### 7.3.3. Cilostazol

Cilostazol is a phosphodiesterase III inhibitor that, in many Asian countries, has been approved in the treatment of secondary non thromboembolic stroke prevention in association with aspirin or clopidogrel, proving effective and not associated with an increased risk of bleeding in treated patients [63,64]. In a recent retrospective study, the authors compared ASA plus clopidogrel (standard therapy) versus cilostazol plus clopidogrel (alternative therapy) in patients who underwent percutaneous coronary intervention. Out of 613 total patients, 205 reported aspirin intolerance and received alternative therapy with cilostazol. The authors reported no significant difference in MACEs between the two groups (*p* = 0.12), with a trend toward reduction in bleeding events in the cilostazol group (0.49% vs. 2.7%, *p* = 0.063) [65]. These results need to be confirmed by further, randomized studies.

#### 7.3.4. Aspirin-Free PCI in Emergent Settings

In cases of STE-ACS, very high risk NSTE-ACS, or any other condition that requires immediate PCI, a practical option is to perform the procedure without ASA administration and using a P2Y12 inhibitor (including cangrelor during the procedure, and the potent P2Y12 receptor inhibitors ticagrelor or prasugrel afterwards) plus an intravenous continuous infusion of a glycoprotein IIb/IIIa inhibitor. This will allow for performing the PCI under an appropriate state of platelets inhibition, and will also guarantee the possibility to start an ASA desensitization protocol immediately after the procedure [7,53].

#### 7.3.5. P2Y12 Inhibitor Monotherapy

For patients with chronic coronary syndromes (CCS), long term single antiplatelet therapy is recommended after at least one month of DAPT after PCI. The recently published 2024 guidelines on CCS recommend clopidogrel 75 mg daily as a safe and effective alternative to aspirin monotherapy in patients with a prior myocardial infarction or remote PCI (class of recommendation I-A). Alternatively, for selected patients at high ischaemic risk and without high bleeding risk who were initially treated with ticagrelor-based DAPT, ticagrelor long-term monotherapy (90 mg twice a day) may be considered (class of recommendation IIb-C) [66,67]. For such stable patients, in addition to P2Y12 receptor inhibitor monotherapy, in case of suspected aspirin hypersensitivity, we suggest to perform a LDAC to confirm whether they are truly hypersensitive, in order to be prepared to perform ASA desensitization in case of need of future PCI/ACS. Also, after an ACS, the guidelines allow a P2Y12 receptor inhibitor monotherapy after at least 1 month of DAPT (class of recommendation IIb-B) [68].

Nevertheless, whilst 2019 CCS guidelines allowed prasugrel or ticagrelor monotherapy after PCI if DAPT could not be used because of aspirin intolerance, [69] current CCS and ACS guidelines do not provide recommendations for practical management of patients with aspirin hypersensitivity in the early phase (1–3 months) after a PCI or an ACS, when DAPT is recommended. Few evidences are in fact available regarding a complete aspirin-free SAPT strategy in the first month after PCI/ACS, and these come from small or not globally representative trials [70,71].

The first month of antiplatelet therapy in patients with aspirin hypersensitivity and who cannot (or failed to) undergo desensitization remains therefore an unsolved challenge to date.

#### 7.3.6. Combination of Oral Anticoagulant Therapy and a P2Y12 Inhibitor

The concomitant use of an oral anticoagulant (OAC) plus a P2Y12 inhibitor (usually clopidogrel) may represent a further alternative. For patients with ACS/PCI without a concomitant indication for OAC, evidence for the use of low dose (2.5 mg daily) rivaroxaban plus clopidogrel/ticagrelor (the so-called dual pathway inhibition) versus standard DAPT (aspirin plus clopidogrel/ticagrelor) are scarce [72], and come from a single randomized clinical trial that investigated the safety of this approach, which was not powered to assess superiority in terms of efficacy (ischaemic events) [73]. This approach is also corroborated by the evidence, coming from pharmacodynamic studies, that the use of a P2Y12 inhibitor (clopidogrel or ticagrelor) in combination with low-dose of rivaroxaban is associated with similar effects on platelet-mediated global thrombogenicity but reduced thrombin generation, compared to DAPT [74,75]. No further dual pathway inhibition regimens have been however investigated in clinical trials.

An aspirin-free approach was instead investigated in several trials in ACS/PCI patients with a concomitant indication for OAC, showing a superior safety and similar efficacy of the dual (P2Y12 inhibitor plus standard dose of a direct OAC, DOAC) vs. triple (DAPT plus DOAC) antithrombotic therapy [76,77].

The combination of a P2Y12 inhibitor (mostly clopidogrel) with a DOAC at standard dose may therefore represent a practical approach to overcome the problem of the first phase of antithrombotic therapy after ACS or PCI in patients with aspirin hypersensitivity

## 8. Acute Treatment of Aspirin Hypersensitivity

Acute treatment of ASA hypersensitivity is similar to that of any other hypersensitivity reaction. First line therapies for ASA hypersensitivity acute reactions are corticosteroids or antihistamines. If symptoms are life-threatening or coherent with a diagnosis of anaphylaxis, the first choice is adrenaline. If symptoms are mainly respiratory-related (NERD), first choice therapies are leukotriene receptor antagonists and bronchodilators. In a specific group of NERD patients who experienced chronic rhinosinusitis symptoms and nasal polyps, a reduction in the recurrence of polyps and in the necessity for sinus surgery was noted following chronic aspirin desensitization [9]. Regardless of the hypersensitivity reaction type, a supplementary oral therapy with corticosteroids or antihistamines based on the kind and duration of symptoms could be considered [78].

If a patient experienced a severe reaction to ASA or anaphylaxis, it is mandatory to stop ASA administration and to avoid it in the future. In this case, both LDAC and ASA desensitization protocols are contraindicated since the risk of a recurrence is too high. Thus, alternative reperfusion strategies or antithrombotic therapies, as already mentioned in Section 7.3, should be chosen. If other NSAIDs are chosen as antithrombotic therapies, patients should be informed on potentially cross-reactive NSAIDs as well as with a list of safe alternative NSAIDs, and the safety of the administration of other NSAIDs should be confirmed by tolerance tests [9]. If non-NSAIDs are chosen as alternative antithrombotic therapies, it should be remembered that avoiding NSAIDs will not improve the clinical course and the possible progression of underlying NERD, NECD, or urticaria/angioedema [9].

## 9. Practical Recommendations

A practical algorithm to manage a patient with CAD and suspected ASA hypersensitivity is depicted in Figure 2 (central figure).

The first step is to collect a precise and detailed medical history. This will allow to differentiate ASA intolerance from possible hypersensitivity. In the former case (i.e., patients reporting retrosternal pyrosis, nausea, dyspepsia, other gastrointestinal symptoms or bleeding), ASA administration will be possible, with some precautious such as reducing the dose to 75 mg/day and adding a proton pump inhibitor. In case of persistent suspicion of ASA hypersensitivity, the management differs based on the clinical setting.

In an acute setting, if an emergent PCI is required (STE-ACS or very high risk NSTE-ACS), the procedure can be performed administering a P2Y12 inhibitor (including cangrelor during the procedure) plus an intravenous continuous infusion of a glycoprotein IIb/IIIa inhibitor, and the desensitization protocol is to be started as soon as possible after the procedure. In case of an urgent procedure (non-emergent setting), a practical approach is to tailor the decision-making to the possibility to consult the allergologist and to perform desensitization within the following 24–48 h.

In the chronic setting, for all patients in who ASA hypersensitivity is still suspected based on clinical history, we suggest performing a LDAC to identify those patients who tolerate antiplatelet aspirin dose and that can, therefore, safely undergo PCI with standard DAPT. In case of positive LDAC (i.e., truly hypersensitive patients), desensitization is to be performed before PCI, unless contraindicated. Table 2 provides an example of two LDAC and desensitization protocols, used in previous studies and that can be reproduced in clinical practice [51]. Lastly, for those patients with previous history of anaphylactic shock, in who ASA challenge and desensitization are contraindicated, or for those few in whom the latter was unsuccessful, alternative strategies are to be considered, including the use of alternative antiplatelet agents (i.e., indobufen 100 mg twice a day), antithrombotic regimens (clopidogrel plus full dose of a DOAC or clopidogrel/ticagrelor plus Rivaroxaban 2.5 mg twice a day), or even revascularization strategy (i.e., coronary artery bypass graft).

## 10. Conclusions

Aspirin hypersensitivity represents a significant clinical challenge in patients with CAD who require aspirin as part of an antithrombotic treatment. The percentage of patients reporting aspirin hypersensitivity is in fact high, although the real prevalence is low in general population, making it crucial to identify those patients who really are hypersensitive. The mechanisms underlying aspirin hypersensitivity, moreover, are not fully understood, and further studies are needed to better understand the pathophysiology beneath this condition.

Despite these challenges, and depending on the clinical setting, multiple strategies are available to allow aspirin administration in patients with CAD and reporting drug hypersensitivity, and suspicion of this condition should not automatically deter the use of aspirin. The low-dose ASA challenge and desensitization are in fact safe and effective strategies to overcome this condition, although still underused in clinical practice. For those few patients who cannot (or fail to) undergo ASA desensitization, further approaches may be also considered, including the use of alternative antiplatelet agents or antithrombotic regimens.

## Figures and Tables

**Figure 1 biomolecules-14-01329-f001:**
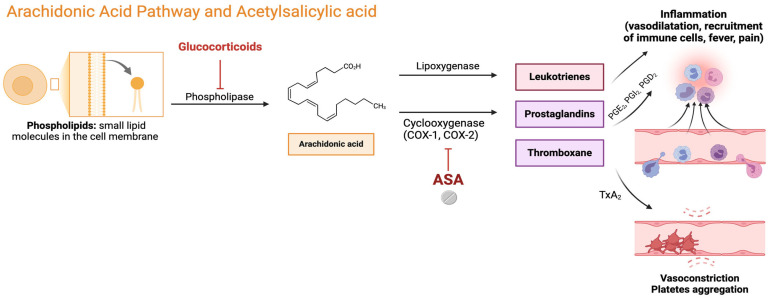
Arachidonic acid pathway in inflammation. The Figure shows the arachidonic acid pathway and the formation of its metabolites (leukotrienes, prostaglandins, and thromboxane) through cyclooxygenase (COX-1 and COX-2) action. Acetylsalicylic acid, blocking COXs, inhibits metabolites effects and exerts anti-inflammatory and antiplatelet function. ASA: acetylsalicylic acid; PG: prostaglandin; TxA_2_: thromboxane A2.

**Figure 2 biomolecules-14-01329-f002:**
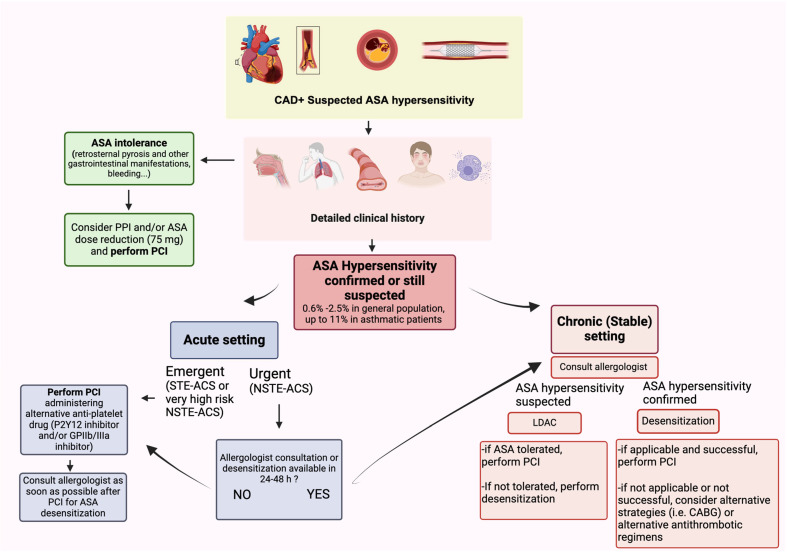
ASA hypersensitivity: diagnostic and therapeutic algorithm. When ASA hypersensitivity is suspected in a patient with atherosclerotic coronary artery disease, a diagnostic workout to confirm or exclude hypersensitivity is mandatory in order to perform percutaneous coronary interventions safely. A detailed clinical history can distinguish ASA intolerance from hypersensitivity signs and symptoms, above all in the presence of comorbidities (chronic rhinosinusitis, nasal polyps, asthma, food allergy). If ASA hypersensitivity is confirmed (or still suspected) in an emergency setting, PCI should be performed by administering an alternative antiplatelet drug. If patient conditions allow for 24–48 h waiting and allergologist consultation is available, LDAC and/or desensitization should be performed. ASA: acetylsalicylic acid; CAD: coronary artery disease; LDAC: low-dose ASA challenge; PCI: percutaneous coronary interventions.

**Table 1 biomolecules-14-01329-t001:** ASA hypersensitivity classification, mechanisms, and manifestations.

	Non-Immunologically Mediated (Cross-Reactive)			Immunologically Mediated Selective (IgE O T-Cell Mediated)	
				Acute	Delayed
	NERD	NECD	NIUA	SNIUAA	SNIDR
Associated conditions	Chronic upper airways disease (CRS, NP, asthma)	Chronic spontaneous (idiopathic) urticaria	None	None	Unknown
Symptoms	Dyspnea, wheezing, rhinorrhea, nasal congestion, laryngeal angioedema	Urticaria/angioedema	Urticaria/angioedema	From mild pruritis, skin swelling, and urticaria up to severe reactions (angioedema, laryngeal obstruction) and anaphylaxis (wheezing, dyspnea, and bronchospasm)	From delayed urticaria and eczema up to more severe reaction (DRESS, Stevens–Johnson syndromes, and toxic epidermal necrolysis)
Time (from exposure)	1–3 h	1 h	<6 h?	<1 h	24–48 h
ASA desensitization	Yes	Usually, no response	Yes	Yes, except anaphylaxis	Unknown

Abbreviations: ASA: acetylsalicylic acid; CRS: chronic rhinosinusitis; DRESS: drug reaction with eosinophilia and systemic symptoms; NERD: NSAID-exacerbated respiratory tract disease; NECD: NSAID-exacerbated cutaneous disease; NIUA: NSAID-induced urticaria/angioedema; NP: nasal polyposis; SNIDR: selective NSAID-induced delayed type HS reactions; SNIUAA: selective NSAID-induced urticaria, angioedema, and/or anaphylaxis,?: the data is still not clear based on available evidence.

**Table 2 biomolecules-14-01329-t002:** LDAC and desensitization protocols. Adapted from Cortellini G. et al. [52,54]. * Continue observation for 1–2 h after procedure.

Ldac Protocol			
Dissolve 288 mg of lysine acetylsalicylate (L-ASA), equivalent to 160 mg of ASA, in 16 mL of water (concentration = 10 mg/mL)			
Time (m)	Milliliter of L-ASA solution	ASA dose (mg)	Cumulative dose (mg)
0	0 (placebo)	0	0
20	1	10	10
65	2.5	25	35
110	2.5	25	60
155 *	5	50	110
Desensitization Protocol			
Dissolve 288 mg of lysine acetylsalicylate (L-ASA), equivalent to 160 mg of ASA, in 16 mL of water (concentration = 10 mg/mL)			
Time (m)	Milliliter of L-ASA solution	ASA dose (mg)	Cumulative dose (mg)
0	0 (placebo)	0	0
20	0.01	0.1	0.1
40	0.1	1	1.1
60	0.2	2	3.1
80	0.3	3	6.1
100	0.4	4	10.1
120	0.5	5	15.1
140	1	10	25.1
180	1.5	15	40.1
240	2.5	25	65.1
300*	3.5	35	100.1

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
