# Peer review of "Aspirin Hypersensitivity in Patients with Coronary Artery Disease: An Updated Review and Practical Recommendations"

_biomolecules, 2024, doi:10.3390/biom14101329_

Round 1
Reviewer 1 Report
Comments and Suggestions for Authors
This is an interesting review which is useful for clinicians of various specialities. I have a few points which need to be addressed:
1. If there are types of food which may trigger aspirin sensitivity please mention them as you referred in the text
2. The current dose of aspirin in the United Kingdom is 75mg while in the text a dose of 100mg is referred to. Did you talk about sensitivity with a 100mg dose only? What about sensitivity with dose 75mg?
3. There are other antiplatelets which is worth mentioning such as Prasugrel and Ticagrelor
Author Response
Comment 1: If there are types of food which may trigger aspirin sensitivity please mention them as you referred in the text
Response: We thank the referee for the suggestion. Manuscript has been amended in order to better clarify the issue. In particular, in section “Nonallergic (pharmacological) reactions” we added a detailed description of the involved foods (please find the changes highlighted in the uploaded text).
Comment 2: The current dose of aspirin in the United Kingdom is 75mg while in the text a dose of 100mg is referred to. Did you talk about sensitivity with a 100mg dose only? What about sensitivity with dose 75mg?
Response: We thank the referee for the observation. In response to this comment, we better explained this aspect in various part of the text, and in particular we specified that:
- as antiplatelet agent the maintenance dose is 75-100 mg/daily (“Introduction”)
- Aspirin hypersensitivity can be classified based on its pathophysiological mechanism into nonallergic (non-immunologically mediated, based on a pharmacological mechanism) and allergic (immunologically mediated). If the former may be dose-dependent, depending on the COX-1 inhibition, the latter are usually dose independent, so the mechanism described in those cases apply to both 75 mg and 100 mg doses (“Definition”).
- Studies have demonstrated that the challenge with ASA at the cumulative dose of 110 mg is a simple, reliable, and a cost-effective procedure. LDAC is performed by administering 10, 25, 25, and 50 mg of ASA every 45 min up to a cumulative dose of 110 mg. The test is obviously interrupted when a positive response is observed. So, thanks to LDAC, a dose of ASA 75 mg could be taken into account in patients with ASA hypersensitivity, who did not tolerate a greater (i.e. 100 mg or more) dose (“Low-dose ASA challenge (LDAC)”).
- The most recent aspirin desensitization protocols in patient with coronary artery disease induce tolerance to aspirin dosages up to 100 mg or even more, with effectiveness around 100% (and the same efficacy is obviously expected for the 75 mg dose).
Comment 3: There are other antiplatelets which is worth mentioning such as Prasugrel and Ticagrelor.
Response: We thank the referee for the comment. In response, we mentioned the possible use of potent P2Y12 receptor inhibitors in sections “7.3.4. Aspirin-free PCI in emergent settings” and “7.3.5. P2Y12 inhibitor monotherapy” (please find the changes highlighted in the uploaded manuscript).
Reviewer 2 Report
Comments and Suggestions for Authors
I have read this manuscript with great pleasure. It presents the topic of aspirin hypersensitivity in a very clear and easy-to read way starting from the pathomechanism to diagnosing and management strategy. Most of the facts are well -known among allergologists. However, the authors aim to present the topic to cardiologists together with outcomes of current clinical trials on alternative antiplatelet therapies.
My suggestion for the authors would be to use the same nomenclature throughout the manuscript in terms of phenotypes of ASA hypersensitivity (NERD, NECD, NIUA, SNIUAA, SNIDR).
Comments on the Quality of English LanguageEnglish language is quite fine. I have found some minor mistakes only.
Author Response
Comment: My suggestion for the authors would be to use the same nomenclature throughout the manuscript in terms of phenotypes of ASA hypersensitivity (NERD, NECD, NIUA, SNIUAA, SNIDR).
Response: we thank the reviewer for the comment. We edited the nomenclature in the entire text and tables as suggested (NERD, NECD, NIUA, SNIUAA, SNIDR), specifying it in section “Definition”. The changes have been highlighted in the uploaded files.